# Setting the Terms for Zoonotic Diseases: Effective Communication for Research, Conservation, and Public Policy

**DOI:** 10.3390/v13071356

**Published:** 2021-07-13

**Authors:** Julie Teresa Shapiro, Luis Víquez-R, Stefania Leopardi, Amanda Vicente-Santos, Ian H. Mendenhall, Winifred F. Frick, Rebekah C. Kading, Rodrigo A. Medellín, Paul Racey, Tigga Kingston

**Affiliations:** 1Department of Life Sciences, Ben-Gurion University of the Negev, Be’er Sheva 8410501, Israel; 2Institute of Evolutionary Ecology and Conservation Genomics, University of Ulm, 89069 Ulm, Germany; luis.viquez@alumni.uni-ulm.de; 3Laboratory of Emerging Viral Zoonoses, Istituto Zooprofilattico Sperimentale delle Venezie, 35020 Legnaro, Italy; sleopardi@izsvenezie.it; 4Graduate Program in Population Biology, Ecology and Evolution, Emory University, Atlanta, GA 30322, USA; amanda.vicente@emory.edu; 5Programme in Emerging Infectious Diseases, Duke-NUS Medical School, Singapore 169857, Singapore; ian.mendenhall@duke-nus.edu.sg; 6Bat Conservation International, Austin, TX 78746, USA; wfrick@batcon.org; 7Department of Ecology and Evolution, University of California, Santa Cruz, CA 95060, USA; 8Department of Microbiology, Immunology and Pathology, Colorado State University, Fort Collins, CO 80523, USA; rebekah.kading@colostate.edu; 9Institute of Ecology, National Autonomous University of Mexico (UNAM), Mexico City 04510, Mexico; medellin@iecologia.unam.mx; 10The Centre for Ecology and Conservation, University of Exeter, Exeter TR10 9FE, UK; p.a.racey@exeter.ac.uk; 11Department of Biological Sciences, Texas Tech University, Lubbock, TX 79409, USA

**Keywords:** bats, Chiroptera, conservation, emerging infectious diseases, public health, science communication, zoonoses

## Abstract

Many of the world’s most pressing issues, such as the emergence of zoonotic diseases, can only be addressed through interdisciplinary research. However, the findings of interdisciplinary research are susceptible to miscommunication among both professional and non-professional audiences due to differences in training, language, experience, and understanding. Such miscommunication contributes to the misunderstanding of key concepts or processes and hinders the development of effective research agendas and public policy. These misunderstandings can also provoke unnecessary fear in the public and have devastating effects for wildlife conservation. For example, inaccurate communication and subsequent misunderstanding of the potential associations between certain bats and zoonoses has led to persecution of diverse bats worldwide and even government calls to cull them. Here, we identify four types of miscommunication driven by the use of terminology regarding bats and the emergence of zoonotic diseases that we have categorized based on their root causes: (1) incorrect or overly broad use of terms; (2) terms that have unstable usage within a discipline, or different usages among disciplines; (3) terms that are used correctly but spark incorrect inferences about biological processes or significance in the audience; (4) incorrect inference drawn from the evidence presented. We illustrate each type of miscommunication with commonly misused or misinterpreted terms, providing a definition, caveats and common misconceptions, and suggest alternatives as appropriate. While we focus on terms specific to bats and disease ecology, we present a more general framework for addressing miscommunication that can be applied to other topics and disciplines to facilitate more effective research, problem-solving, and public policy.

## 1. Introduction

Effectively communicating complex scientific findings to diverse professional audiences, as well as policy makers and the public, can be very challenging. Yet, failure to do so results in misperceptions and misunderstandings that can hamper the progress of science and have significant consequences for the development of public policy. Zoonoses are diseases that originate in animals, either wild or domestic. Zoonotic diseases, especially those from wildlife, garner great attention from policy makers and the wider public because of their impacts on global public health and biodiversity conservation. As the emergence of SARS-CoV-2 from an unknown reservoir leading to the COVID-19 pandemic highlights, effective communication is central to all levels of response, from understanding the origins and evolution of the pathogen to developing policies that can mitigate its spread in human populations. However, communication is challenging because understanding the ecology and emergence of these diseases requires interdisciplinary research, including for example wildlife biologists, ecologists, virologists, microbiologists, evolutionary biologists, and epidemiologists, among others [1].

Bats are an order of mammals encompassing over 1400 species that vary greatly in their life history and ecology [2]. Recently, some bat species have been associated with certain zoonotic diseases. The communication surrounding these findings has exemplified the challenges outlined above. For example, “bats” in general are frequently described as reservoirs of various emerging human pathogens in both the scientific literature and popular press, although specific associations are restricted to only one or a few of the over 1430 known species [2]. Further, in some cases there is ample evidence indicating that particular bat species are in fact reservoirs of certain pathogens, such as Nipah virus [3], while for other viruses, such as ebolaviruses, the natural reservoir is not well-defined [4,5]. In yet other cases, microbes detected in some bat species are similar to those causing human disease but this indicates only an evolutionary relationship, not transmission of a pathogen currently circulating in bat populations [6]. These nuances are often misunderstood by the public [7], which directly threatens bat conservation through increased negative attitudes towards bats, as well as persecution, eviction and even government proposals to cull populations [7,8,9,10].

Effective communication relies on intentional transmission by the sender of unambiguous messages that are received and understood by the target audience [11]. Over time, scientific disciplines have developed specific terms and norms to minimize ambiguity in communication. Understanding of these terms and conventions is typically gained through instruction and many years of practice. Experts and academic societies also play an important role in standardizing terminology within a discipline [12,13]. However, scientific discovery and advances mean that the lexicon of science is constantly expanding, presenting opportunities for miscommunication even among experts within a discipline and even more so between experts across disciplines or subdisciplines, due to diversity of training, language, and philosophies. Complex, interdisciplinary issues of contemporary interest to the general public are especially vulnerable to message corruption because they involve even more diverse audiences with different domains of knowledge and levels of understanding.

In communication about zoonotic diseases and bats we identified four recurrent types of miscommunication that are profoundly impacting inference and thereby interdisciplinary communication, public opinion, and, potentially, policy (Table 1). They arise from problems in messaging by the sender, understanding of the receiver, or both. The types of miscommunication are:*Incorrect or overly broad use of terms*—words in which the sender is unaware of the accepted definition of a term in the field(s) that coined it and so uses the term erroneously, or without the level of evidence that would support that use, e.g., bats as the reservoir of MERS-CoV or SARS-CoV-2. This perpetuates misinformation and misunderstanding in audiences, whether they are other researchers or the general public.Terms that have *unstable usage* within a discipline (e.g., bat origin), or different usages among disciplines (e.g., endemic, vector), leading to confusion or misunderstanding in audiences.Terms that are used correctly but spark *incorrect inferences about biological processes* or significance in the audience (e.g., spillover, novel).*Incorrect inference from the evidence* presented. This may be due to the fact that the audience (and occasionally the messenger) is unfamiliar with the methodologies generating the evidence (e.g., serological evidence, phylogenetic evidence).

We note that some terms (e.g., intermediate host) may be misused in multiple ways that can fall into more than one category.

Here we present a short glossary of terms that illustrate these four types of miscommunication that we believe have caused the most confusion and disruption to effective, accurate communication regarding zoonotic diseases and bats. We have organized the presentation of these terms into four broad groups: actors or entities, such as “pathogen” or “reservoir”; processes or events, such as “spillover” or “spillback”; descriptors, such as “endemic” or “novel”; methodologies, such as “serology” or “phylogeny.” For each term, we note which problem type(s) it falls under and provide a definition, examples of correct use, and an overview of common misconceptions, misuse, and caveats. We recommend alternatives where appropriate. While we have focused on terms that we find have been commonly misused with regard to bats during the COVID-19 pandemic, misuse of these terms and the types of miscommunication that they exemplify are also applicable to other species, situations, and disciplines.

## 2. Glossary of Terms

**a.** 
**Entities**
*(i.)* *Pathogen* (Type 1)

Definition: A pathogen is a microbe that either causes damage in a susceptible host or has the potential to do so [14,15,16]. The microbe in question can be a bacterium, virus, fungus, or eukaryote, such as *Plasmodium*. While pathogenicity, the ability of a microbe to cause damage to a host, was originally conceptualized as intrinsic to the microbe, it has become increasingly clear that damage is due to the interaction between a microbe and a specific host, and may be caused by the pathogen (e.g., ability to kill cells, release toxins), the host immunological response (e.g., cytokine storm or infection-induced autoimmunity), or a combination of both [14,15]. Thus, the pathogenicity of any given microbe depends not only on its intrinsic properties, but on the host being considered [14]. For example, Zaire ebolavirus, Sudan ebolavirus, Taï forest ebolavirus, and Bundibugyo ebolavirus are pathogenic to humans, non-human primates, and ungulates, while Reston ebolavirus is pathogenic to non-human primates but not humans, and none of these ebolaviruses appear pathogenic to bats [17]. Similarly, the variation in pathogenicity of different *Rickettsia* species within the spotted fever group to humans has recently been attributed to the different ways in which these very similar bacterial species manipulate host cell function [18]. It is important to note that within a single population or species, a microbe may be pathogenic to some individuals but not to others, depending on genetic factors, immune function, or other host characteristics [14]. In addition, the environment in which the hosts and the pathogens interact might affect the outcome of the infection [19].

Appropriate use: The term “pathogen” should be used to describe microbes that cause damage or disease in a host. Whenever “pathogen” is used, the host species in which the microbe may cause damage should be specified.

Caveats and common misconceptions or misuse: “Pathogen” is sometimes used loosely as a catch-all term to describe microbes that have been isolated or detected in samples, without determining whether they cause damage or disease in hosts (which often requires the examination of tissues and cells by experts [20]) or specifying the hosts in which the microbes cause damage. For example, microbial species or genera that include some pathogenic strains, such as *E. coli* or *Pseudomonas* sp., are sometimes broadly and incorrectly referred to as pathogens. Even viruses, which need a host’s cellular machinery to reproduce [21], may be non-pathogenic to their hosts (e.g., [22,23,24]). Further, while modern sequencing approaches provide powerful new tools for microbe discovery and form the basis of “pathogen discovery” initiatives and pipelines, it is not always clear how the pathogenic nature of newly-discovered microbes is being assessed, which would require specific criteria [25], especially when the relationship is based on genotypes.

While authors are generally vigilant in referring to the newly described viruses from such studies as “potentially pathogenic”, the leap from microbe to pathogen may also occur in media reporting. For example, in a recent preprint, Hul et al. [26] describe a coronavirus detected in archival samples collected from *Rhinolophus shameli* in 2010 in Cambodia with 92.6% similarity to SARS-CoV-2. While the authors clearly note that, “further risk assessment is needed to understand the host range (including humans) and pathogenesis associated with this novel sublineage” ([26], p. 4) subsequent reporting in other outlets described this virus as a “pathogen” [27].

Recommended alternatives: If a microbe has not been shown to cause damage or disease in a host, it could be described either generally as a “microbe” or “microorganism” or preferably more specifically as a type of microbe, such as bacteria, virus, fungus, or protozoan, when applicable. A microbe could be described as a “potential” or “putative” pathogen if it shares virulence factors with well-characterized pathogens or in light of sufficient molecular or clinical evidence. For example, while the pathogenic potential of Bombali ebolavirus is not yet known, it has still been described as a “potential pathogen” because it shows genetic similarities to both ebolavirus species that are pathogenic to humans and Reston ebolavirus, which is pathogenic only to non-human primates [28]. Bombali ebolavirus can also bind and enter human cells, but this finding does not necessarily mean the virus will ultimately cause disease in people [28].

*(ii.)* *Reservoir or Reservoir Host* (Type 1)

Definition: A reservoir is a population, species or community (assemblage of different species in a given geographic area) in which a microorganism naturally occurs and is indefinitely maintained [29,30]. Some zoonotic pathogens, particularly bacterial pathogens, may also have environmental reservoirs [31]. Microorganisms with multiple reservoir species may be indefinitely maintained across the community of species even if they are not always present in each individual reservoir species. In the reservoir species or community, the microorganism may cause either asymptomatic infection or disease and this may vary among individuals [32]. A pathogen may also be more genetically diverse in its reservoir host than in other taxa [33], in part because the pathogen is endemic to reservoir hosts.

Some authors further specify that a reservoir should be defined in the context of another species of interest, called a target species [32]. While the concept of a target species may be helpful in some contexts, it is not a requirement for defining a reservoir. Usually target species are humans, domestic animals or a specific wild species of interest, such as an endangered species (for example, African wild dogs (*Lycaon pictus*) are a target species for rabies spillover from domestic dogs (*Canis lupus familiaris*) [34,35]). Transmission of the microbe from the reservoir species to the target species is generally the focus of concern if the microbe is highly transmissible or pathogenic to the target species. In this particular case, the reservoir is defined as an animal that maintains the microbe in nature and can transmit it to the target species, thus acting as a source for reintroduction even after successful control [32]. However, this may be an overly restrictive definition [30] in several senses. First, a population in which a microorganism is maintained would arguably be its reservoir even if transmission to a target population has never occurred, especially because it is impossible to know if transmission will occur in the future. For example, Shimoni virus, which belongs to the genus *Lyssavirus*, was first isolated from the leaf-nosed bat *Hipposideros commersoni* in 2009 [36]. Although there has been no known spillover of this virus to any specific target populations, studies have tried to establish whether *H. commersoni* or other bat species are its reservoir, the population in which the virus is maintained [37]. (Note that since the study was published, the entire *H. commersoni* group has been reassigned to the genus *Macronycteris* [38] and it is unclear whether original records refer to *M. gigas* or *M. vittatus*). This knowledge is important for understanding the ecology and dynamics of Shimoni virus even if spillover to other target populations does not occur, and it will be essential if it does. This underscores the point that a population would have to serve as a reservoir in which a microbe circulates prior to spillover to a target population. For example, dromedary camels (*Camelus dromedarius*) appear to have been a reservoir for MERS-CoV since at least the 1980s [39], long before spillover to humans was detected in 2012 [40,41] and certain populations also appear to be reservoirs in regions where spillover does not occur [42]. 

Appropriate use: We advise caution when asserting that a species or community is the reservoir of a microbe. Establishing that a particular population or species is a reservoir for a microorganism is difficult and requires multiple lines of evidence to demonstrate susceptibility to infection, subsequent pathogen shedding, long-term maintenance and circulation in the population of a given species, potentially including field, laboratory, and epidemiological studies [43,44].

When the term reservoir is used in relation to a target species, the reservoir species should be found infected with the same strains occurring in the target species and should be proven responsible for the transmission of the microorganism [32]. The pathogen in question should have high genetic and functional similarity, for example using the same receptor, infecting the same cell types, or releasing the same toxins, in the reservoir as in the target species or population into which it has spilled over [43]. In addition, the designation of a species as a reservoir should denote the specific taxonomic nomenclature or common name that refers to an individual species. For example, the common name of an animal, such as “jaguar” or “wolf”, might refer to a single species or a population of the species within a given geographic region. However, in many other cases, a single common name such as “bat” (see below), “bird”, or “fish” could refer to many different species grouped, respectively, in the same order, class, or even a paraphyletic group, with limited biological meaning. If no common name at the species level exists, a broader common name may be paired with the scientific name, e.g., “the fruit fly species *Drosophila bifurca*”.

Caveats and common misconceptions or misuse: Species should not be called a reservoir when genetically related microorganisms are detected in them, especially if detection is based upon a single gene. Phylogenetic inferences regarding common ancestry should also not be used to assign reservoir status for pathogens [43]. For example, bats have been incorrectly described as the reservoir of SARS-CoV-1, SARS-CoV-2, and MERS-CoV [45,46]. Bats are the evolutionary reservoir of alpha- and beta-coronaviruses and can correctly be referred to as the reservoir for diverse SARS-like coronaviruses [47]. However, the human pathogens SARS-CoV-1 and SARS-CoV-2 have not been detected in bats. Thus far, phylogenetically-related viruses that are found in bats have not been shown to have the same infective capability in humans, and divergence between SARS-CoV-1, SARS-CoV-2, or MERS-CoV, and a common ancestor in bats likely occurred decades before these viruses infected humans [45,46,48,49]. However, our knowledge of the diversity of coronaviruses in bats, including SARS-like coronaviruses, is still poor and further research could identify more closely related SARS-like coronaviruses or possibly indicate a direct spillover of SARS-CoV-1 or SARS-CoV-2 from bats but current evidence cannot support such an assertion.

Further, the sporadic identification of microorganisms in a wildlife host is not direct evidence for its role as a reservoir [44]. Indeed, such infections could be spillover events and the investigated species may be unable to maintain the infectious cycle, thus representing a dead-end host. This could be the case, for example, of dogs (*Canis lupus familiaris*) found positive for SARS-CoV-2 during the COVID-19 pandemic that, up to now, seem to become infected by their owners, but do not play a larger epidemiological role [50]. Thus, longitudinal investigations rather than cross sectional studies are necessary to investigate the ecology of infectious diseases [51], and specifically to identify reservoirs. Experimental infections to investigate the dynamics of infections can support the reservoir status of a species, but this is often not possible for wild species due to the lack of taxonomically-relevant captive colonies for many microbe-host pairings.

In some cases, bats have been presumed to be the reservoir of pathogens when evidence remains limited and the conditions for determining a reservoir (see above) have not been fully met. For example, bats, and in particular bats in the paleotropical plant-visiting family Pteropodidae, (commonly referred to as Old World fruit bats, Megachiroptera, or megabats—although we advise against using the latter two terms as they are based on outdated taxonomy [52]), are often referred to as the reservoir of ebolaviruses, including ebolavirus species that cause disease in humans [53]. Exposure to ebolaviruses has been detected by PCR or serology in bat species belonging to the families Pteropodidae, Hipposideridae, Rhinolophidae, Molossidae, Vespertilionidae, and Miniopteridae across Africa and Asia [28,53,54,55,56,57,58,59,60], but PCR prevalence and seroprevalence are generally low and ephemeral [4] and species in the genus *Ebolavirus* have not been isolated from bats (or any other animal), a gold standard to identify a host (although this may be due generally low levels of virus circulation in individual bats, hindering isolation). While numerous bat species appear to be regularly exposed to ebolaviruses and certainly play a role in the viruses’ ecology and dynamics, the natural reservoirs are currently unknown [5,61]. 

Recommended alternatives: “Reservoir” should only be used in cases that fulfill the criteria outlined above. When the ancestor of a pathogen is suspected to have originated in a particular taxon, this taxon should be referred to clearly as an ancestral reservoir, the reservoir of a common ancestor, or the reservoir of related microbes, which should be as specific as possible (Figure 1). For example, bats may be referred to as the reservoir of SARS-like coronaviruses or the ancestral reservoir of alpha- and beta-coronaviruses [47]. On the other hand, if a microorganism is detected in a species once or on rare occasions, the species could be referred to as an “incidental” [30] or “natural” host, or simply as a “host” to clearly distinguish it from a documented reservoir (Figure 1). A reservoir of an infectious agent may also be referred to as a “source population” to reflect the connectivity between the reservoir and target population to which the infectious agent is transferred in the case of cross-species transmission [32].

*(iii.)* *Intermediate host* (Type 2, Type 1)

Definition: The term “intermediate host” was first used to describe multi-host parasites, such as helminths or protozoa that grow or complete a part of their life cycle in one or more hosts (the intermediate(s)) before infecting their final, definitive hosts, where their sexual phase occurs [62,63]. 

“Intermediate host” is frequently used in reference to viruses but there is no standard definition in this context. This term has generally been used to indicate species, often domestic animals or livestock, that facilitate the spillover of microbes from wildlife to humans. Hosts that have been described as “intermediate” are often domesticated or peridomestic (e.g., pigs, camels) and therefore presumed to have higher contact rates with humans than the wild reservoir species or community, so that they act as a bridge between the two. In other cases, the virus can be amplified or might undergo selection for improved transmission in the intermediate host, further favoring spillover [64]. However, the use of “intermediate host” in virology differs from the parasitological context as transmission to the intermediate host is not always a necessary step for the biological cycle of the microorganism (in fact this type of usage is more akin to the term “paratenic hosts” in traditional parasitology [65,66]). For example, populations of at least four species of *Pteropus* bats are reservoirs of Nipah virus (*P. hypomenalus*, *P. lylei*, *P. medius*, *P. vampyrus*) [3,67,68,69] but it was first described as a human pathogen after its initial transmission from infected pigs (*Sus scrofa domesticus*) [70,71]. Transmission from bats without the involvement of pigs is also possible under certain circumstances and has been confirmed in Bangladesh, where transmission occurs via human consumption of collected palm sap contaminated with urine or saliva of *Pteropus medius* [72,73].

Appropriate use: “Intermediate host” is appropriately used when referring to hosts that are infected by a multi-host parasite in its immature form before infecting the definitive host [62,63]. We recommend against using “intermediate host” for other types of pathogens, including viruses. See Caveats and Recommended alternatives for further discussion.

Caveats: We caution against using the term “intermediate host” in the context of zoonotic viruses, in large part because there may be confusion and inconsistency between the categorization of host species as “intermediate hosts” or reservoir hosts. This is partially due to temporal ambiguity: a host described as “intermediate” for a virus can become a reservoir host if the virus is able to establish and maintain an infectious cycle in it through transmission to susceptible individuals. This new reservoir can become the primary source for human infection due to higher contact rates with humans compared to the original reservoir species. The microbe may also undergo selection in the “intermediate” host that favors either human transmissibility or pathogenicity. In this case the microbe may become sufficiently differentiated from its ancestor and adapted to the “intermediate host” that the “intermediate host” then becomes the reservoir of this new microbial species or strain that has diverged from the ancestral microbe found in the ancestral reservoir (see “Bat origin” for further discussion). 

For example, the critical role of dromedary camels (*Camelus dromedarius*) in contributing to the emergence of MERS-CoV as a human disease is widely recognized [74]. While the host of the ancestor of MERS-CoV is still unknown, current evidence indicates that it was likely a bat species belonging to the family Vespertilionidae or Emballonuridae [75,76]. However, MERS-CoV appears to have been maintained in camel populations since at least the 1980s [39] and is now widely endemic in these animals even where human outbreaks do not occur [42,77,78]. Thus, regardless of its ancestral wildlife source, dromedary camels, which at one point may have merely been “intermediate” hosts, have effectively become the reservoir host of MERS-CoV, and source of human outbreaks [46,49].

In some cases, there is simply not enough evidence to definitively determine whether a species played the role of an “intermediate host.” For example, palm civets (generally referring to either *Paradoxurus hermaphroditus* or *Paguma larvata*) and raccoon dogs (*Nyctereutes procyonoides*) are frequently described in both the scientific literature and popular media as “intermediate hosts” of SARS-CoV-1 although it is still unclear if these species were involved in spillover to humans [79].

Recommended alternatives: Due to the differences in the transmission and “life cycles” of viruses compared to animal parasites as well as the potential ambiguities in separating reservoir hosts from intermediate hosts outlined above, we recommend avoiding the term “intermediate hosts” in the context of zoonotic viruses. The term “amplifying” host [80,81] or “bridge” or “bridging” host better describes the functional role that species other than the reservoir hosts might have in the process of spillover to humans [81].

*(iv.)* *Vector* (Type 2)

Definition: The definition of “vector” varies considerably across disciplines. In the context of infectious disease transmission, the various definitions and contexts for the term “vector” have been recently reviewed by Wilson et al. [82]. One of the most commonly-utilized and traditional definitions of a vector comes from a medical and veterinary perspective, and refers to hematophagous arthropods (e.g., flies, ticks, mites) that carry and transmit an infectious agent between hosts [82,83]. This definition includes biological as well as mechanical roles in transmission (e.g., house flies transmitting bacteria [84]) in which the arthropod serves as a vehicle to transfer the infectious agent between vertebrate hosts. Public health agencies such as the US Centers for Disease Control and Prevention Division and the European Centre for Disease Prevention and Control subscribe to this definition of a vector [85,86]. 

Appropriate use: In our view, defining vectors as arthropods that feed or land on vertebrates, invertebrates, or plants and thus carry and transmit pathogens to these hosts, is the most accepted and useful definition when dealing with vector-borne diseases, since it covers relevant aspects of vector biology (e.g., life cycles and response to the environment). Another advantage of this definition is the distinction of a vector from an intermediate host, since these arthropods are micropredators that improve their fitness at the cost of the host (even if insignificantly) [87], while definitive and intermediate host fitness are not directly affected by each other.

Caveats and common misconceptions or misuse: The term “vector” has been loosely applied to cases in which organisms other than arthropods are capable of infecting a person or other animals to the infectious agent. Examples of this usage include rodents and hantaviruses [88] and dogs (*Canis lupus familiaris*) and rabies virus [89]. The WHO also has an anthropocentric definition of vector: any living organism that can transmit infectious pathogens between humans, or from animals to humans [90]. Using this definition, any animal that transmits a microbe to humans is considered a vector, while an organism that transmits the microbe between non-human species, would not be considered a vector. This narrow definition would of course exclude all vector-borne transmission driving the enzootic or sylvatic cycles of myriad vector-borne pathogens. Bats have also been referred to as animal vectors of disease [91], to which we posit that bats would potentially serve as the reservoir, and transmission of the infectious agent to human or other animals would occur via a direct as opposed to a vector-borne route.

Recommended alternatives: We suggest avoiding “vector” when referring to non-arthropod hosts of pathogens and instead stating that the pathogen is “transmitted by” the species in question. Vertebrates may be referred to as a host or reservoir of the pathogen as appropriate (see Reservoir). Specifying “invertebrate vector” may also aid in avoiding misunderstandings.

**b.** 
**Processes and Events**
*(i.)* *Spillover* (Type 3)

Definition: A spillover is an event in which a microbe is transmitted from one species (usually the reservoir but potentially an amplifying or bridge host) to a novel, susceptible species, establishing infection in this individual new host [92,93,94] (Figure 1). Spillover may be limited to a single individual or the first case of spillover may be the index case for a larger outbreak [94].

Appropriate use: In nature, spillover events are not as rare as some may believe. In general, zoonotic diseases in humans are often results of spillover events from a natural reservoir. For example, every case of rabies in humans is a result of spillover from a mammalian host, usually through a bite or scratch from an infected host [95]. Furthermore, serological evidence suggests that human exposure to known and potential zoonotic pathogens might be even higher than indicated by outbreaks of disease, as some spillover appears to result in abortive infections or sporadic, isolated cases of small clusters that remain undetected [96]. Over 60% of emerging diseases in humans are zoonotic and 71% of these are the result of spillover from the wildlife [97], sometimes passing through an amplifying or bridge host, as shown for Nipah virus [80,98]. In any of these cases, for successful spillover to occur, a microorganism needs several concomitant conditions resulting in strict contact between the pathogens shed by an infected animal host and a susceptible recipient host. 

Caveats and common misconceptions or misuse: For the general public, “spillover” may be immediately linked to a zoonotic outbreak, epidemic, or even pandemic, an idea reinforced by news headlines [99], book titles such as David Quammen’s popular *Spillover: Animal Infections and the Next Human Pandemic* (2012), and references in films such as *Contagion* (2011) or *Outbreak* (1995). It is important to emphasize that spillover does not necessarily lead to a larger scale outbreak and could result in only a single case of infection in humans [94]. For example, each spillover of rabies to humans results in a single case with no onward transmission or larger outbreak [95]. Outbreaks resulting from spillover events are only possible when the new host is able to perpetuate the infection with intraspecific transmission. Even then, only in relatively rare cases does spillover lead to an epidemic (a rapid increase in disease prevalence in a region) or even rarer yet a pandemic (an epidemic spread over multiple continents) [100]. When epidemics or pandemics do occur, this is not generally due to repeated spillover events from the reservoir host, but to the high viral fitness of the pathogen in the new host, which often translates into high viral titers, contagiousness, and transmission between humans, as exemplified by the ongoing COVID-19 pandemic [101] or the 2013–2014 Ebola epidemic [102].

Recommended alternatives: The term spillover should be only used referring to the initial cross-species transmission event of a microbe. It should not be used as a proxy for an outbreak or the existence of a wildlife zoonosis more generally. 

*(ii.)* *Spillback* (Type 1)

Definition: Spillback generally indicates a specific case of spillover (see Spillover) when microbes spill from a new host back to the original host [103,104] (Figure 1). Kelly et al. [103] also include the transmission of macroparasites from a new host back to the original host in their definition of spillback. 

Appropriate use/examples: Spillback was first used to describe the spillover of *Brucella* from wild ungulates to cattle (*Bos taurus*) around Yellowstone National Park. Cattle had likely transmitted *Brucella* to wild ungulates, who then became a source of infection for cattle where they came in contact [105,106]. Other examples include the spillback of bovine tuberculosis to cattle from white-tailed deer (*Odocoileus virginianus*) in North America [107] or from badgers (*Meles meles*) in the UK [108] and the recent case of SARS-CoV-2 spillback from mink (formerly *Neovison vison*, recently revised as *Neogale vison* [109]) to humans [110]. 

Caveats and common misconceptions or misuse: Recently spillback has been used to generally describe the spillover of zoonotic pathogens from humans to any animal species [111], such as the potential spillover of SARS-CoV-2 from humans to North American bat species [112] or deer mice (*Peromyscus maniculatus*) [113]. Because the original use of spillback referred to transmission from a new host back to the original host, such wording implies that the species to which spillback occurs is the original reservoir host. This is not the case with SARS-CoV-2 and bats or other wildlife in North America—the original host of SARS-CoV-2 remains unknown and is most likely from China or a neighboring country [101]. 

Recommended alternatives: Spillback should refer to spillover events from a new host back into the original host species. In all other cases of transmission of microbes from humans to wildlife, “spillover” or “cross-species transmission” is preferable. “Pathogen pollution” is another term that can be used to refer generally to anthropogenic introduction of pathogens to new areas and hosts [114].

*(iii.)* *Mutation* (Type 3)

Definition: A mutation is a change in a nucleotide sequence (DNA or RNA), such as substitution, deletion, or insertion. These changes may affect a single base-pair or many base-pairs. Mutations may be beneficial, neutral, or deleterious. 

Caveats and common misconceptions or misuse: The meaning and significance of a “mutation” can easily be misunderstood by the public. This may be due in part to the way the term is used in science fiction and popular culture, where it is often associated with a massive transformation or the acquisition of dangerous characteristics. However, mutations in the natural world are often neutral or deleterious to the organism or may not impact its phenotype [115,116]. In viruses, mutations are normal and relatively frequent. However, during the COVID-19 pandemic for example, there has been extensive reporting on mutations in SARS-CoV-2, often focusing or speculating on their effects on transmissibility or virulence [115]. In addition, the many SARS-CoV-2 variants whose mutations do not appear to affect transmissibility or virulence [117] have received less media attention, again reinforcing the misperception that mutations in general make viruses more dangerous.

Appropriate use: While the use of “mutation” in scientific literature is correct, we urge caution when communicating results regarding mutations to the media and the public, avoiding sensationalizing and limiting speculation. Extensive experimental and epidemiological data is needed to determine the effect of mutations on the transmissibility or virulence of a virus [115]. Until such evidence is available, the possible effects of mutations should not be overstated.

**c.** 
**Descriptors:**
*(i.)* *Novel/New* (Type 3)

Definition: Novel or new is applied to anything that is being observed or described for the first time, such as a novel virus or a new animal species.

Caveats and common misconceptions or misuse: In the context of a “new” species of plant or animal, it is generally understood by biologists that the species itself is not new, but it is being newly described by scientists (although such species may be well-known to indigenous or other local people).

However, in the case of microbes or pathogens, using the terms “new” or “novel” could be misconstrued to mean that they did not exist before and have only recently evolved. In most cases, these “new” or “novel” microbes are simply newly described by science. There can be further confusion when organisms are truly novel, having only recently evolved, such as the emerging, novel variants of SARS-CoV-2 [118,119,120]. 

Recommended alternatives: We recommend using “newly described”, “newly identified”, or “newly discovered” as more accurate phrases when referring to microbes or pathogens that are observed or described for the first time. The term novel should only be used if there is evidence to suggest the microbe has (very) recently evolved.

*(ii.)* *Endemic* (Type 2)

Definition: Endemic has a distinctly different meaning in general ecology compared to epidemiology, and by extension disease ecology. In ecology and biogeography, endemic means a species is native to an area, usually with a restricted geographic distribution. For example, island endemic species are species that occur on a specified island or islands and nowhere else. This usage was coined by Charles Darwin in *The Origin of Species* [121].

In contrast, in epidemiology, endemic refers either to a disease that is persistently prevalent in a given geographic area or a geographic area in which a disease is persistently prevalent with no sudden changes within a given time period [122,123,124]. For example, yellow fever is endemic to sub-Saharan Africa and sub-Saharan Africa is an endemic area for yellow fever. A disease or geographic area with persistently high prevalence may be considered “hyperendemic” [124]. This use of “endemic” dates to Hippocrates in the fourth century BCE [123] and exists in contrast to an epidemic, which is an increase of a disease in a given geographic area, usually within a short time-frame [123,124]. A disease does not have to be native to a geographic area to be considered endemic—it may be introduced to a region and become endemic once it is established. For example, yellow fever is considered endemic to South America, even though it was likely introduced from West Africa [125]. Epidemiological endemicity may vary by temporal scale. Diseases, such as dengue, may be endemic over multiyear time periods but seasonally epidemic [126].

For diseases circulating only in animals, the equivalent terms are enzootic and epizootic [127], although “endemic” and “epidemic” are widely used in this context as well.

Appropriate use: Clearly, the use of endemic in both the ecological and epidemiological sense is correct. In order to avoid any misinterpretation or ambiguity, we suggest that authors simply state clearly the definition or sense in which they are using “endemic.” Further clarity of epidemiological use can be conferred by specifying geographical extent and relative time scales.

*(iii.)* *Bat origin* (Type 2, Type 3)

Definition: There is no standard definition of “bat origin.” This term has been used to describe both microbes isolated from or detected in bats [128,129] and microbes, such as SARS-CoV-2, that may have descended from a common ancestor in bats, but which themselves have not yet been detected in bats [6,112,130].

Caveats and common misconceptions or misuse: In the case of SARS-CoV-2 “bat origin” has clearly been used as shorthand to illustrate the evolutionary relationship between ancestral bat coronaviruses or sister groups of bat coronaviruses and SARS-CoV-2. However, this use is easily misinterpreted by both scientists who are not experts in virology or phylogenetics, and the public, who may incorrectly understand such phrasing to mean a pathogen that is transmitted directly from bats to humans, that is, that they are the origin of the pandemic [7]. Rather, “bat origin” is meant to indicate that the ancestor of SARS-CoV-2 likely originated in bats of the genus *Rhinolophus* [6]. This confusion is compounded by the alternative use of the phrase to mean a microbe that has been isolated from or detected in bats [128,129]. 

Recommended alternatives: For pathogens whose ancestor came from bats, but which have not themselves been detected in bats, we suggest instead using the phrase “virus descended from an ancestor found in bats,” “diverged from a bat virus”, or “evolutionary bat origin.” This type of phrasing emphasizes the evolution of viruses from an ancestor found in bats as well as the time for such divergence to occur, while also differentiating the current virus we observe from related species found in bats.

*(iv.)* *Bats* (Type 1) 

Definition: The term “bats” is a common name that can refer to the entire order Chiroptera, several species within the order, or more than one individual of a single species. The order comprises over 1400 known species [2], encompassing great ecological, biological, and physiological diversity, with the more distantly related clades of bats separated by more than 50 million years of evolutionary divergence [52,131]. Bat species differ in what they eat (fruit, nectar, leaves, insects and other arthropods, blood, fish, vertebrates) [132,133,134], how far they travel nightly or seasonally (<1 km to a migration in 1000 km) [135,136,137], their social structure (solitary, small groups, dynamic fission-fusion systems, large colonies in the 1000s to millions) [138], and where they roost (e.g., exposed in large trees, beneath understory leaves, tree hollows, caves, or domestic dwellings) [139]. Life history characteristics are equally varied; for example, longevity can range from a few years to four decades and reproduction may be seasonally monoestrus to asynchronously polyoestrus [140,141]. Differences in ecological and life history traits are likely related to physiology and immunology, although this is still poorly understood [142]. 

Bat species also vary in the types and closeness of contact they have with humans. Some species are completely intolerant of any human disturbance and thus are restricted to unmodified habitats, while others are synanthropic, meaning they share human dwellings, even in dense urban areas [143,144,145]. In parts of the world, some bat species are routinely exploited for food and medicine [146,147]. The ecological and biological diversity of bat species thus confers great variability among species and populations as hosts of viruses, viral transmission dynamics, and the human-bat interface. 

Caveats and common misconceptions or misuse: “Bats” is widely used as a synonym for the entire order Chiroptera. Although some papers may report the taxonomic and ecological diversity of bats, they rarely recognize the significance of this diversity for species-specific trait combinations that influence both the disease dynamics themselves, as well as the size and riskiness of the human-bat interface. For example, an elusive, solitary rainforest species likely presents less potential risk than one roosting in large mixed-species aggregations in caves regularly visited by people [148]. In particular, introductory statements of many studies lump bats together using statements such as: “Bats are reservoirs for deadly viruses,” which implies all 1400+ species are reservoirs (and see complications with use of reservoir). This language is then often repeated in popular media, such as a *Wall Street Journal* article from 9 April 2020 entitled “*The bats behind the pandemic*” followed by the subtitle: “*From Ebola to Covid-19, many of the deadliest viruses to emerge in recent years have the same animal source*” [149]. While there are still many uncertainties about from which animal species ebolaviruses and SARS-CoV-2 originate (see Reservoir), this refers to bats belonging to different species, genera, and families within the order Chiroptera as “the same animal source.” 

This “lumping” of species extends to inference from studies as well. Most field and laboratory studies can only focus on a small subset of bat species, but findings are often implicitly extended across taxa and contexts. Similarly, if “bats” is being used to denote multiple conspecifics, it may imply that all individuals within populations are currently infected by a virus, even when prevalence is low. López-Baucells et al. [8] found that 70% of studies on viruses in bats did not report the proportion of infected bats and 62% did not describe a potential transmission pathway from bats to people. 

Recommended alternatives: While all bats have wings, only some bats (species or individuals) currently host a particular virus, especially one that is pathogenic to humans or other animals, and even fewer bat species have been demonstrated as reservoirs (see definition above). Whenever possible, it is best to specify the family, genus, or species of bats being discussed and report the prevalence of microbes detected in each bat species in order to avoid implicitly extended findings to all bat species. Common names may also be included, particularly for communication with the public. Efforts should be made to incorporate the ecology of bat species and any specific contact pathways they might have with humans when discussing the results of disease ecology studies in bats [8].

**d.** 
**Methodologies:**
*(i.)* *Serology* (Type 4)

Definition: Serology is the detection of antibodies in the blood serum of an animal induced by an infection with a particular microorganism, which provides evidence for past exposure to that microorganism [150].

Appropriate use and interpretation: Serology overcomes several challenges related to the direct detection of microbes in wildlife, including bats, and is widely used to study the ecology of wildlife diseases. Compared to infectious agents, which are difficult to isolate or detect and soon cleared by the host immune response, antibodies last longer in the individual and can be found in the blood regardless of the organs targeted by the infection, the infectious status of the animal, or the occurrence of shedding at the moment of sampling [150]. Serology is particularly useful for longitudinal studies analyzing the dynamics of viral transmission through changes in the proportion of seropositive individuals over time [151]. This can identify phenological or environmental factors that drive the viral shedding pulses thought to underpin spillover, as shown for Marburg virus in the bat species *Rousettus aegyptiacus* [152]. Seroconversion of experimentally-challenged animals also is one indicator used to confirm susceptibility to infection (e.g., [113]) 

In addition, antibodies elicited by a microbe can cross-react with other microbes in the same genus or family that show antigenic similarity [153]. This information is still important for the study of viruses, especially those such as lyssaviruses [154] or filoviruses [155], that are all considered pathogenic, or likely pathogenic, to humans or other animals and, thus have the same impact on public health regardless of the specific viral species that is circulating. 

While this cross-reactivity makes it challenging to discriminate between the circulation of related microbes, such as viruses in the family *Flaviviridae* [156], it represents an opportunity for the study of wildlife diseases, because it allows for the use of existing serological assays outside their original scope to investigate the circulation of pathogens before their actual isolation or molecular description, including undiscovered pathogens related to those for which the assay was designed [153]. For example, numerous studies reported serological evidence of bat exposure to henipaviruses, filoviruses and lyssaviruses across the Old World before their isolation or molecular identification [57,153,157,158,159,160,161,162,163,164,165]. 

While serological findings alone do not constitute sufficient evidence to resolve the role of a species as reservoir hosts for the human pathogens targeted and used in the assays, these surveys provide strong evidence when coupled with genomic detection and are critical for describing and understanding the exposure history of bats to different groups of viruses [153,163,166].

Caveats and common misconceptions or misuse: While the detection of antibodies can provide useful information on the spread and dynamics of microorganisms in wildlife, it is crucial to acknowledge that serology only provides evidence for past exposure and cannot be used as a proxy for infection or disease [150]. For example, abortive rabies infections that elicit the production of neutralizing antibodies but do not result in disease have been observed in humans, bats, dogs, and livestock [167,168,169]. This is likely to occur with other pathogens as well. On the other hand, seronegativity may not always indicate a lack of exposure or even current infection. For example, cattle (*Bos taurus*) actively infected with pathogenic *Leptospira* can still test seronegative [170,171]. In addition, serological data derived from a single point in time cannot discriminate between an outbreak in a population resulting from cross-species transmission to an incidental host or the persistence of the microorganisms in the reservoir host [32]. Finally, the antibody decay rates in wildlife species, such as bats, may be poorly known.

Another pitfall of the use of serological assays in wild animals is the likelihood for cross-reactivity between the target microbe and other related infectious agents, known and unknown. This means that a serological test can be positive even if the individual has never come in contact with the microbe of interest due to antigenic similarity [153]. For example, 17 different lyssaviruses have been described in bats [172]. Based on the results of passive surveillance, each of these species seems to be associated with one or a few related bat species [154]. However, antibodies neutralizing, for example, European bat lyssavirus-1 have been found in a wide variety of hosts from different genera [162,173,174]. Currently, it is not possible to determine whether European bat lyssavirus-1 has a much broader host range than expected or if these results are due to an unexplored diversity of cross-reacting lyssaviruses in European bat species. 

Finally, the use of serology for the study of microbes in wildlife may be limited by: the unknown specificity and potentially poor accuracy of the test used; the validation and standardization of the tests in different host species, including true positive controls; and the uncertainty in defining the cut-off values used to differentiate between positive and negative individuals [150]. In addition, different laboratories might produce very different results depending on the viral antigen used, the testing platform (e.g., ELISA or multiplex microsphere assays) and the cut-off value they use, making it challenging to compare results among studies [150]. This is particularly true for bats, which often show lower titers of antibodies compared to other mammalian hosts, sometimes falling above or under the cut-off depending on the researcher’s choices [153,175]. 

Recommended mitigation strategies: As shown, it may be challenging to make certain inferences from results of serological assays, as they are often made based on a number of assumptions that may or may not be fully justified [153]. Among the mitigation strategies available to reduce the error, researchers can: standardize serological approaches in the target species, compare results only within a species, provide justification of the cutoff chosen, and/or criteria for determining positivity, and/or report raw data rather than seroprevalence alone, thus allowing for more meaningful comparisons across studies [176]. Additionally, clarifying the limitations of interpretation for the results of the particular assay is useful, for example by stating viruses that were or were not included in the testing panel for cross-neutralization tests, especially regarding viruses that are endemic in the study area or even associated with the studied species elsewhere [177]. Attempts should be made to confirm serological data, including comparison against alternative assays which may detect antibodies in different ways (binding versus neutralization) and assessing the performance of the assay across populations and laboratories. Longitudinal data across populations and seasons is always preferred over cross-sectional studies in order to investigate whether a focal species is a natural host or an incidental dead-end host for the target microbe. Molecular evidence from at least one individual is usually required to confirm infection and characterize the agent [178].

*(ii.)* *Phylogeny* (Type 4)

Definition: A phylogeny is a hypothesis that is tested to determine the evolutionary history and relatedness of organisms by comparing the similarity of different taxa. Although here we focus on phylogenies reconstructed using molecular data, as they are generally the most relevant for pathogens and disease ecology, we note that phylogenies may also be reconstructed using other types of data, such as morphology [179]. In molecular-based phylogenies a sequence (nucleotide or protein), which may be a gene or genome, is compared to sequences chosen to serve as references [180]. An evolutionary model of the rate of molecular evolution over time is then applied to the sequence dataset and is determined by nucleotide or amino acid substitution patterns. Phylogenetic relationships (evolutionary patterns of relatedness among organisms or viruses) are commonly visualized using phylogenetic trees that graphically illustrate the relationships among and between different groups of interest, which may be individuals, strains, populations, species, or other taxonomic groups. These trees can depict ancestral relationships between organisms.

Phylogenetic trees based on molecular data are usually reconstructed using either a Maximum Likelihood (ML) method [181] or a Bayesian approach [182]. Both methods estimate multiple trees to arrive at either a single tree (ML) or a set of trees (Bayesian analysis) that depict the most likely or probable relationships, based on the input data and a given evolutionary model. Depending on the type of reconstruction, the length of the branches represents either the number of substitutions in genetic sequences or time since divergence, based on whether a substitution rate is known or not. Normally, branch support is provided either by bootstrapping (ML methods) or the posterior probability (Bayesian methods).

Caveats and common misconceptions or misuse: A phylogeny is a working hypothesis because it draws only from the input data and therefore it is only as reliable and accurate as the information used in its reconstruction. Poorly curated alignments and selection of unrepresentative sequences can result in incorrect estimations of ancestral relationships. Incomplete datasets can misrepresent how close or distant a relation is between two adjacent branches. This can be a common issue when newly-described viruses emerge because there are relatively few reference sequences (sequences from related strains or taxa) available for comparison. A paucity of reference sequences, sampling bias, and selecting an inappropriate root sequence often produces phylogenetic trees with unsupported branches, resulting in inconclusive relationships [183]. Further, interpretation of the closest phylogenetic relative of an organism or virus is based on sequences used in the alignment. If new, more closely related sequences are added to the alignment the phylogenetic tree branches and thus the inferred relationships among them may shift (see Figure 2). For example, increased Sarbecovirus (Genus: Betacoronavirus) sequence data from bats has demonstrated the diversity in this group [184] and why SARS-CoV-2 may have a wide host rage [185]. The robust sequencing of SARS-CoV-2 in humans has facilitated the identification of variants that may be of concern and the subsequent diagnostic assays that detect them [186]. It is also important to note that phylogenies only describe the patterns of descent but not the phenotypic or functional similarities between two adjacent taxa [187]. Further, caution should be used when interpreting posterior probability in Bayesian phylogenies because values are nearly always high and do not necessarily reflect the reliability of the tree [188]. However, this can be mitigated by using multiple genetic markers, unbiased data sets, and ensuring an appropriate model is selected [189].

The choice of gene(s) used to build a tree is also important and will affect the results of any phylogenetic analysis. The choice of gene should reflect the question being asked [191]. For viruses, slowly evolving genes (e.g., replicative genes or polymerases), are useful to evaluate relationships between more distantly related groups, while faster evolving genes (e.g., surface proteins) are good markers to study fine-scale evolution of closely related viruses. Coronavirus phylogenetic analysis can be challenging because this virus family undergoes recombination frequently [192]. This occurs when two different coronaviruses replicate in the same host cell, resulting in offspring that have genetic makeup from both “parent” viruses and thus two different evolutionary histories (one from each parent). Misuse of genetic data can provide misleading relationships and wrongly incriminate hosts. One early study on SARS-CoV-2 found codon (three nucleotides that form an amino acid) usage in the virus was most similar to snakes and concluded that these animals were the most likely animal reservoir [193]. However, virus codon usage for a recently emerged virus is unlikely to parallel that of the host; instead comparing a virus to other viruses is more likely to yield reliable insight into likely hosts or ancestral reservoirs [194,195].

Recommended mitigation strategies: Researchers should carefully consider the gene(s) used to create a phylogenetic tree and use genes that are appropriate for the taxa of interest, based on breadth of taxonomic sampling, orthology, and rate of evolution. Orthologous genes are inherited from an ancestral gene and have diverged over time. They are identified either de novo using tree- or graph-based methods or inferred based on comparisons to known reference orthologues [180]. As noted above, for phylogenetic analysis of more distantly related taxa over longer evolutionary time scales, it is necessary to use more conserved (which are often slower evolving) genes, while analysis of more closely related taxa over finer time scales requires genes that evolve faster. Before conducting full genome analysis on viruses that can recombine, it is necessary to perform a breakpoint analysis to ensure that phylogenies represent a single evolutionary history.

Multigene phylogenies should be generated whenever possible, although for some viruses this may not be possible due to the difficulty of designing primers and amplifying regions outside of certain highly conserved regions. Caution should be used when interpreting phylogenies based on short sequences or a limited number of genes. For example, many phylogenies for animal coronaviruses are based on sequences of *RdRp* in the ORF1ab and may only be a few hundred base-pairs long [47]. Global alignments should be made with as many sequences as feasible and downsampled to ensure sufficient representatives are incorporated for each major taxonomic group. Inferences on the relationships between both viruses and viruses and hosts are also hindered by our still-limited knowledge of the full diversity of wildlife viruses [196,197]. A recent study on SARS-related coronaviruses in Thailand showed different relationships between SARS-CoV-2, bat and pangolin sarbecoronaviruses (Genus: Betacoronavirus; Subgenus: Sarbecovirus) depending on which region of the genome was analyzed [198]. This study could not determine whether pangolins are incidental hosts or secondary reservoirs of these coronaviruses, but there are likely undiscovered lineages of sarbecoronaviruses circulating and alignments should be updated to reflect these discoveries. 

## 3. Discussion and Conclusions

In an ideal world, audiences would have greater domain knowledge and more time to review content carefully, report accurately, and thus avoid serious miscommunication. However, because audiences in this context include scientists from other domains, journalists, and the general public, this is not a feasible solution. The onus is therefore on domain experts as messengers to prevent miscommunication by avoiding ambiguity, being aware of points of confusion, sharpening their usage of terminology, and embedding clarification directly into content. At the same time, scientists from other disciplines should be careful about adopting and using words without being clear on their meanings and inference. This is a key component of interdisciplinary research. 

Preventing and minimizing miscommunication regarding bats and zoonotic diseases is important because failure to do so has serious negative consequences for understanding disease dynamics, developing effective mitigation or prevention strategies, and conservation efforts. For example, correctly identifying reservoirs, bridging or amplifying hosts, or origins of pathogens is essential for effectively researching and thus further understanding their ecology, host specificity, transmission dynamics, and pathways of spillover. Similarly, broadly describing “bats” as hosts of pathogens or emerging diseases rather than specifying species, genera, or families implicitly conceptualizes 1400+ different species as essentially the same biological entity, limiting our understanding of the ecology of specific microbes in their specific host species. Since any risks posed by microbes within hosts are species-specific, accurate specification is critical to support inference in disease ecology research. While some studies deliberately choose to use only a few representatives of each taxonomic group to compare patterns among higher taxonomic groups (e.g., selecting a few species from several Orders to compare patterns among Classes of animal), we urge caution in interpreting such results for the reasons outlined above.

Clear communication, including correctly identifying and describing the reservoir, bridging hosts, or origin of a pathogen, is also essential for designing and implementing effective strategies for preventing and mitigating zoonotic disease spillover because these measures largely rely on managing the distribution of reservoirs or other hosts, reducing the prevalence of the pathogen in these species, or minimizing interactions between hosts and target populations [199]. The most effective strategies are tailored to the ecology of reservoir species and the nature of their interactions with target populations, either directly or via bridge hosts [81,200]. For example, direct actions to reduce specific types of contact between pigs or people, and fruit bats in the genus *Pteropus*, such as moving pig sties away from orchards or using skirts to cover date palm sap receptacles has helped prevent the spillover of Nipah virus to pigs and humans [71,72,201]. On the other hand, for pathogens such as SARS-CoV-1 or SARS-CoV-2 whose pathways to spillover are still unknown but appear to have an ancestral origin in bats [6,46,184], continuous, general surveillance of potential hosts and the people that come into close contact with them, as well as educating people living at high-risk interfaces, is likely the most effective prevention strategy.

Finally, miscommunication regarding zoonotic diseases can have harmful effects on conservation. In general, incorrect usage of “reservoir”, “vector” or implying that all individuals of all bat species are dangerous and carry viruses pathogenic to people can provoke fear and negative attitudes towards bats, especially when interpreted or sensationalized in popular media [8,9]. Most recently, the frequent association of bats with COVID-19 has led to increased negative attitude towards bats [7,10], persecution by the public and authorities [202,203], and calls from government officials to cull bats [7]. Even arguably correct—but ambiguous—phrases such as “bat origin”, that summarize complex processes can lead to public confusion. For example, misunderstanding the evolutionary relationship between SARS-CoV-2 and SARS-like coronaviruses detected in bats of the genus *Rhinolophus* alluded to by the term “bat origin” has led people to incorrectly believe that SARS-CoV-2 is directly transmitted from bats to people [7]. Further, because non-specialists and the public in general may be unaware of the taxonomic and ecological diversity of bat species, the frequent use of “bats” as hosts of various zoonotic diseases without mention of specific species, genera, or families in both scientific literature and popular media can lead the public to incorrectly extend reports regarding one species or group of species to others. For example, some residents in Arkansas, USA have associated local bats with COVID-19, leading to negative attitudes towards bats in general even though bats in the genus *Rhinolophus* do not occur in North America and local Arkansas bat species have no link to COVID-19 or SARS-CoV-2 [10]. Incorrect usage of terms in the scientific literature, such as describing the potential transmission of SARS-CoV-2 from humans to bat species in North America as “spillback” [112] can exacerbate this misconception by incorrectly implying that these species are hosts of the virus, in line with the original definition of the term [103].

Some of the terms we have identified lead to miscommunication with potentially lower stakes but nevertheless have implications for research or public understanding of disease ecology. For example, “endemic” has a radically different definition in ecology compared to epidemiology and misunderstandings can hinder interdisciplinary research and collaboration or the interpretation of results by experts from different domains. In another case, incorrect inference of the term “novel” can affect public perception of pathogens and diseases. Believing that a “novel” pathogen is newly evolved, rather than newly described by scientists, may raise a much greater alarm in the public and instill a false sense that many more species of pathogens exist and that they evolve more quickly than they actually do. This may be compounded by using “novel” in this more literal sense, for example, in reference to newly evolved strains of SARS-CoV-2 [118,119,120].

We note that we are not creating new definitions or terms and most definitions that we point to are well-established in disciplines, although in some cases, such as “reservoir” [29,30,32,200], there may be on-going debate and discussion. It is important not to create new terms and definitions if existing ones will serve, as this can easily lead to a multitude of synonyms without widely accepted definitions that are difficult to distinguish. Conversely, attaching new meaning to existing terms, such as “spillback”, can also cause further confusion and should also be avoided. Confirming the meaning of terms in the literature, including verifying the original source of any cited definitions, and acknowledging any disputes in definition where they exist can clarify the use of these terms and avoid miscommunication. 

Our list of terms subject to miscommunication is clearly not exhaustive. We selected those that we believe are causing the most confusion and misunderstanding in the current COVID-19 crisis, and that have downstream consequences for public health policy and practice, and biodiversity conservation. Although we focus on bats, miscommunication surrounds these terms in research and discussions of other taxa involved in emerging infectious diseases (e.g., birds, rodents, livestock). Nonetheless, there are many more terms and contexts causing confusion, and this is only likely to increase with growing appreciation and application of One Health approaches, which by their very nature are at minimum multidisciplinary and at best interdisciplinary. While the amount of interdisciplinary One Health research has grown, there is still a certain level of separation between subdisciplines due to differences in study systems, questions, and methods [204]; a shared vocabulary across disciplines could facilitate further interdisciplinary collaboration. Consequently, we suggest that it is useful for authors to think about the typology of miscommunications we have applied here and identify potential examples of the types in their writing. This can lead to solutions for each type. 

In this vein, we propose solutions for each type of miscommunication (Table 1). For Type 1, incorrect or overly broad use, messengers need to check contemporary definitions and usage, and work to avoid their own assumptions about the definition, particularly if using terms that originated in a discipline different from their own or in a subdiscipline with which they are less familiar. When using Type 2 terms, which may be in flux or differ among disciplines, the messenger needs to clearly indicate which definition they are adopting. Pointing to a reference is not enough, a clear definition should be provided in the body of the text. A laudable example is that of Guth et al. [205] who included a table within their main text defining key terms such as “reservoir host” and “spillover”. Some Type 2 terms, such as “bat origin” that do not have standard definitions should be avoided, particularly in interdisciplinary publications or in material meant for the general public. If retained, the term should be defined at first use with a subclause that clearly states what is meant, for example: “bat origin, by which we mean the virus shares a common ancestor with one found in a specified bat species, so an evolutionary relationship is inferred.” Type 3, addressing incorrect inference about processes by the audience, is particularly challenging. In these cases, it is important not to overstate or sensationalize a term. Rather, authors might add precautionary language, especially in material meant for wider audiences, such as press releases; exaggeration in press releases is highly correlated with exaggeration in subsequent news stories [206]. For example, instead of constantly linking “spillover” to disease outbreaks or the “next pandemic,” authors might instead note that “spillover” means the transmission of a microbe from one species to another and only in some cases leads to a larger outbreak. To avoid Type 4 incorrect inference from the evidence presented, the messenger should be clear about the assumptions and limitations of the methods used. Results should not be exaggerated based on the evidence presented, e.g., referring to a species as the reservoir of a pathogen based only on serological results. Researchers who are less familiar with specific methodologies should collaborate with more experienced experts, especially in interdisciplinary projects or when writing broad review articles that cover evidence from multiple disciplines.

The problems identified above are not exclusive to the written word. The public and policy-makers are generally not reading academic journals, but getting information from sound bites, videos, and interviews disseminated via traditional news outlets and social media [207,208,209], both of which can spread misinformation [210,211]. Within this context, it is especially important that scientists with domain knowledge avoid sound-bite traps by using turns of phrase that concisely capture complex issues, or embed explanations into their responses, for example: “Spillover is when a pathogen is transmitted from an individual of one species to an individual of a different species. Many times, spillover ends with one or a few infections, but sometimes it can lead to a wider outbreak” or “Bat-origin, meaning the virus found in people is similar to a virus found in bats because they shared a common ancestor in the past. But the virus is only transmitted between people, not from bats to people.” Using precise language when discussing zoonotic diseases in any medium or context is key to preventing miscommunication, with positive downstream effects on effective interdisciplinary research, successful prevention and mitigation strategies, and increasing public understanding and awareness while avoiding the negative repercussions on conservation.

## Figures and Tables

**Figure 1 viruses-13-01356-f001:**
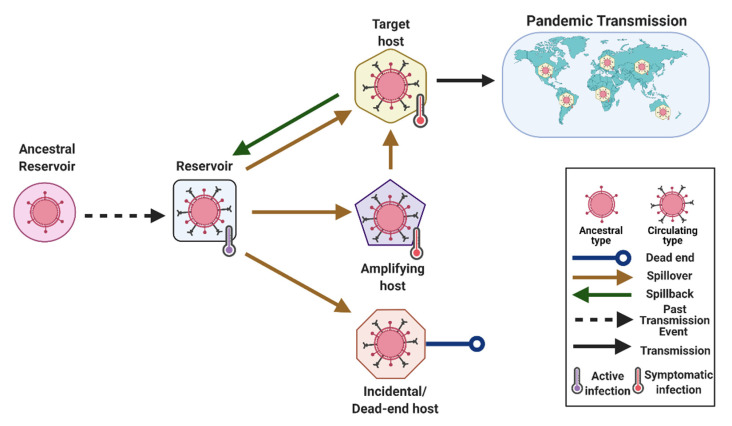
Schematic diagram illustrating some of the key terms defined in this paper. Each colored shape represents a different, generic “species” that serves as a particular type of host for a generic microbe. Arrows indicate different types of transmission, such as spillover and spillback.

**Figure 2 viruses-13-01356-f002:**
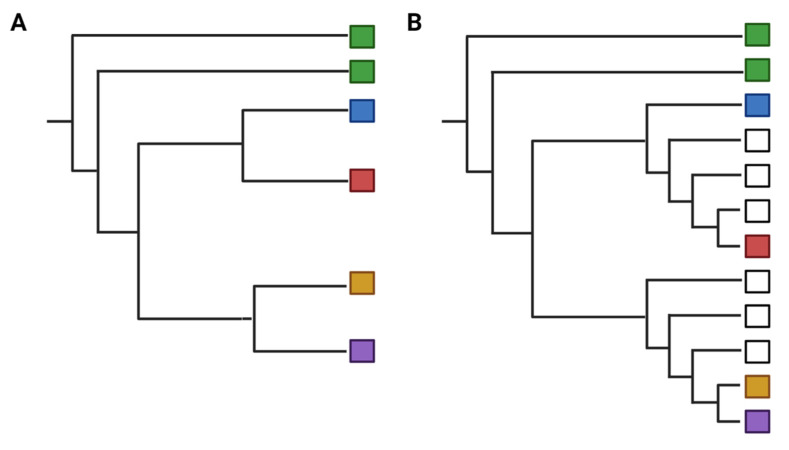
Illustration of how the number of sequences used to build a phylogeny affects the inferred degree of closeness of relationships between taxa. Sister taxa or groups are those that are most closely related to each other and share a unique common ancestor [190]. However, this relationship is sampling-dependent, as shown in the figure. In (**A**), the red and blue taxa are hypothesized to be sister taxa, as are the gold and purple, since they are the closest to each other and share a unique common ancestor. In (**B**), when six new sequences are added to the phylogeny, the gold and purple are still sister taxa; although the red and blue are still in the same clade, they are no longer sister but now appear more distantly related to one another.

**Table 1 viruses-13-01356-t001:** Summary of the main types of miscommunication identified. For each type of miscommunication, we identify the main reason that confusion occurs, whether it originated in the messenger, receiver, or both, and offer possible solutions that can be implemented by either the messenger or receiver. We also list all the terms that we define in this paper and use to exemplify each type of miscommunication. Terms are listed within each miscommunication type in the order they appear in the manuscript.

Miscommunication Type	Source of Confusion	By	Examples	Solution	Implemented by
1	Incorrect or overly broad use	Messenger	Pathogen Reservoir Intermediate host Spillback Bats	Confirm definitions in the literature	Messenger
2	Unstable usage or different usage between disciplines	Messenger and/or Receiver	Intermediate host Vector Endemic Bat origin	Provide definition in text Use more specific alternatives if available	Messenger
3	Incorrect inference about biological process	Receiver	Spillover Mutation Novel Bat origin	Avoid sensationalism Add precautionary language	Messenger and Receiver Messenger
4	Incorrect conclusion from evidence	Messenger and/or Receiver	Serology Phylogeny	Familiarize with different methods	Messenger and Receiver
				Do not overinterpret findings	Messenger and Receiver
				Collaborate with experts	Messenger

## Data Availability

Not applicable.

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
