# Peer review of "Setting the Terms for Zoonotic Diseases: Effective Communication for Research, Conservation, and Public Policy"

_viruses, 2021, doi:10.3390/v13071356_

Round 1

Reviewer 1 Report

This MS deepens into an important miscommunication problem regarding the relationship between the science done regarding this last pandemia (and former epidemias) and its spread to the public (including politicians) and the impact of this miscommunication on wildlife and particularly on bats.

In fact, misconceptions and/or misuse of technical terms for zoonotic diseases are considered as the main factor leading to the worrisome decay in the public perception of bats and their role in ecosystems.  The terms are grouped as entities (pathogen, reservoir), processes (spillover, spillback), descriptors (endemic, novel) and methodologies (serology, phylogeny). 
The terms are also categorized into five types and each term is presented and discussed in detail and alternative definitions are presented and use recommendations.
The paper is very clearly written and is supported by an outstanding collection of most seminar references which is praiseworthy given the width and variety of aspects that the paper has to deal with, touching different fields like epidemiology, virology, public health and bats biology. 

As a result, I consider that this work can be useful for the scientific community and the general public as well. Moreover, I think its arrival in this post-pandemic era is particularly convenient since the effects of these misconceptions on wildlife and particularly on bats perception and conservation can be very damaging.

I have just a few comments and/or suggestions for some of the terms presented in the paper:

-Regarding the miscommunication types, I'm not sure if the type-3 one (use of jargon) is at the same level as the rest, and it may be more convenient to merge this type with type-2 (unstable terms) since one of the problems of using jargon is also its instability and ambiguity according to who is the 'messenger'.

Pathogens: 
I absolutely agree with the overuse of this term and the need of considering the particular host in any mention of a pathogen.

Reservoir:
I'm not sure if the word 'target' projects the same touch of 'directionality' that has in my language. If this was the case also in English and 'target' gives the impression of directionality, I'd try to avoid this term since the 'target species' of a microbe can just appear by chance or in a particular moment or conditions, and of course, it is difficult to predict upfront.
Please update the name of Hipposideros gygas as Meganycteris gygas, see Foley et al., 2017 (https://doi.org/10.3161/15081109ACC2017.19.1.001).

Intermediate host:
It's a nice example of the variety of meanings that the same world can have if used by virologists or parasitologists. Still, the concept of a temporary host to get closer to humans and/or undergo a rapid evolution towards increasing the transmissibility and pathogenicity to humans and therefore its chance of a successful spillover to humans seems quite real and valid as an epidemiologic concept. I doubt as a consequence that the specialists will accept the proposed change to 'bridge species'.

Vector:
Although there are indeed different interpretations for vectors I don't see the harm in extending the idea of its use to any organism (besides arthropods) capable of transmitting or infecting an infectious agent. The loosen concept is quite straightforward and I'm afraid will continue to be widely used.

Bat origin:
It may be good to emphasize here (as it's done in the Discussion) the deep difference between 'origin of the pandemic' (totally unknown so far) and the 'evolutionary origin of the virus SarsCov2' which seems was a bat-hosted coronavirus.

Bats:
It seems to me that the misuse of this term fits probably more into the type-1 category (incorrect or overly use) and 'bats' do not refer to a particular biological process by themselves as it's stated for the type-4 category. Anyhow, I agree that part of the problem stems from the fact that 'bats' are commonly considered just as a single homogeneous group of animals.

Serology and Phylogeny:
The concerns and limitations about making inferences using these two particular methodologies are generally applicable and relevant to any situation or problem to which they could be applied. 

Particularly regarding the Phylogeny term, I'd like to point a few things despite the risk of being maybe too picky:
Definition: As it is said, a Phylogeny is a hypothesis; but a phylogeny doesn't 'TEST' the relationships. Instead, it shows hypothetical relationships to BE TESTED later on.

 The word CONSENSUS should be avoided in L 711 since it applies only to BA. Maximum Likelihood (ML) produces by definition a single most likely tree whereas Bayesian analysis (BA) produces a set of most likely trees -given the data- (Hall, B. G. 2011. Phylogenetic trees made easy: A how-to manual). I'd rather just say in L 710: Both methods estimate multiple trees to arrive at a single tree (ML) or set of trees (BA) that depict the most likely or probably relationship based on the input data and A GIVEN EVOLUTIONARY MODEL. Don't forget this last addendum.

Posterior probabilities (PP) may be 'nearly always high'  (L for genomic data, but this certainly doesn't apply when using single markers (still widely used) and this statement should be specified in L 732.

As I said the problems of biassed or insufficient sampling, reconstruction methods, evolution rates, markers, etc. are common to any phylogenetic reconstruction and it would be probably better to summarize them and rather focus more on the consequences and caveats of the inferences on the SarsCov evolution.

Please, correct the misspelling and change 'sites' for 'sties' at the beginning of L 820.

Finally, I want to congratulate the authors for the enormous amount of work put in the writing and bibliographic revision put on the preparation of this important paper.

Reviewer 2 Report

It's a good point to do this kind of research, the authors will save many animal life, this paper is in good shape, and I suggest to accept it as it is!

Author Response

We thank the reviewer for their positive comments.
As they have suggested to accept the manuscript as is, we have no specific comments to respond to.